# ON THE CONVERGENCE OF LoRA-BASED FEDERATED LEARNING: A UNIFIED ANALYSIS OF AGGREGATION-BROADCAST OPERATORS

## ABSTRACT

Federated Learning (FL) enables collaborative model training across decentralized data sources while preserving data privacy. However, the increasing scale of Machine Learning (ML) models poses significant communication and computation challenges in FL. Low-Rank Adaptation (LoRA) has recently been integrated into FL as a Parameter-Efficient Fine-Tuning (PEFT) strategy, substantially lowering communication costs by transmitting only a small set of trainable parameters. Nevertheless, how to aggregate LoRA-updated local models on the server remains a critical and understudied problem. This paper presents a comprehensive theoretical analysis of LoRA-based FL frameworks. We first classify existing aggregation schemes into two main categories: Sum-Product (SP) and Product-Sum (PS). We then introduce the Aggregation-Broadcast Operator (ABO) as a general class encompassing all aggregation-broadcast methods. Any method in this class ensures local or global convergence as long as the corresponding Weak or Strong Convergence Condition is satisfied. In particular, we prove that the SP and PS aggregation methods satisfy the weak and strong convergence conditions, respectively, but differ in their ability to achieve the optimal convergence rate. Moreover, we conducted extensive experiments on standard open datasets to verify our theoretical findings.

**AI Acknowledgment:** We acknowledge that AI tools were employed to assist in paper writing and polishing the text to improve readability.

## 1 INTRODUCTION

Federated Learning (FL) has emerged as a promising framework for training machine learning models across decentralized data sources while preserving data privacyMcMahan et al. (2017); Kairouz et al. (2021). However, the growing complexity of modern deep neural networks poses significant challenges for communication efficiency and resource-constrained devices, which are central challenges in FL Deng et al. (2020). Low-Rank Adaptation (LoRA), a parameter-efficient fine-tuning technique initially proposed for large-scale language models Hu et al. (2022a), has recently attracted increasing interest in FL due to its ability to reduce communication overhead by updating only a small subset of trainable parameters. The core idea of LoRA is to constrain the weight update on the model by a low-rank decomposition:

$$W' = W_0 + \Delta W, \quad \Delta W = BA, \tag{1}$$

Where $B \in \mathbb{R}^{d \times r}$ and $A \in \mathbb{R}^{r \times n}$, with $r \ll d$. By training only $B$ and $A$, LoRA significantly reduces the number of trainable parameters while maintaining model performance. Using LoRA in an FL setting is an effective and resource-efficient strategy. The global objective of LoRA in FL can be expressed as:

$$\arg \min_{A,B} \frac{1}{m} \sum_{i=1}^{m} \mathbb{E}_{(x,y) \sim P_{XY}^{(i)}} \left[ \mathcal{L} \left( W_0 + AB; (x,y) \right) \right]. \tag{2}$$

where $m$ denotes the number of clients, and $(x,y) \sim P_{XY}^{(i)}$ indicates that the local data of client $i$ follows the distribution $P_{XY}^{(i)}$. By leveraging LoRA adapters, clients can fine-tune large foundation

models with minimal computational overhead. Because only the low-rank adapter matrices need to be communicated with the central server, this approach greatly reduces communication overhead. Compared to full-parameter fine-tuning, LoRA offers a more scalable and efficient solution for improving model performance in collaborative learning environments.

Despite its advantages, integrating LoRA into FL introduces new challenges, particularly in how the locally updated low-rank parameters are aggregated on the server side Yang et al. (2025). Unlike traditional FL, where full model weights or gradients are averaged directly, LoRA-based training requires the design of specialized aggregation strategies that respect the low-rank structure. In recent studies, multiple LoRA aggregation methods have been proposed, which we broadly classify into the following categories in this paper:

**Sum-Product-Type (SP) Aggregation Method.** This method is referred to as the SP method throughout the paper. Such a method is also referred to as the ideal aggregation method, as it shares the same form as FedAvgMcMahan et al. (2017). Its aggregation form is as follows:

$$\Delta W = \frac{1}{m} \sum_{i=1}^{m} B_i A_i \tag{3}$$

This aggregation form can unify several recent methods. For example, FlexLoRABai et al. (2024) was the first to aggregate local models to the server by Eq. (3), and then broadcast by Singular Value Decomposition(SVD). FedIT Zhang et al. (2024a) uploads locally fine-tuned LoRA parameters from each client, which are then aggregated on the server using FedAvg to update the global model. FLoRA Wang et al. (2024), a stacking-based LoRA aggregation method, further improves this process by reducing the impact of noise during aggregation.

**Product-Sum-Type (PS) Aggregation Method.** This method is referred to as the PS aggregation method throughout the paper. Its aggregation form is as follows:

$$\Delta W = (\frac{1}{m} \sum_{i=1}^{m} B_i)(\frac{1}{m} \sum_{i=1}^{m} A_i) \tag{4}$$

This form encompasses several existing methods, such as Zero-PaddingCho et al. (2023) and RBLA Chen et al. (2024a) for Heterogeneous LoRA aggregation. FFA-LoRASun et al. (2024), which freezes the LoRA matrix $A_i = A_0$ and only aggregates the LoRA Matrix $B_i$. Moreover, RoLoRAChen et al. (2024b) employed an alternating form, aggregating only $B_i$ in odd rounds and $A_i$ in even rounds. Different from these, FedSA-LoRAGuo et al. (2025) updates and learn both $B_i$ and $A_i$, but only the $A_i$ matrices are shared for aggregation to learn general knowledge, and saves $B_i$ locally for capturing client-specific knowledge.

**Other Aggregation Method.** Some aggregation methods cannot be easily categorized as either SP or PS type aggregation methods, such as FedIncQin & Li (2024), which proposed a clustering-based aggregation method, enabling more fine-grained and adaptive aggregation. FedEx-LoRASinghal et al. (2024) and LoRA-fairBian et al. (2024) introduce correction terms during aggregation to make the results closer to the SP aggregation(ideal aggregation). CoLRNguyen et al. (2024) adopts a hybrid aggregation strategy by locally learning matrix A while globally sharing matrix B through server-side decomposition. LoRA-A$^2$Koo et al. (2024) employs alternating minimization with adaptive rank selection to reduce communication costs by focusing on the most important LoRA ranks.

**Motivation** Although the two methods mentioned above are now widely used, their underlying mechanisms remain unclear, as we still face the following questions:

- The SP aggregation method is referred to as the ideal aggregation methodYang et al. (2025); Guo et al. (2025), but is it truly the fastest in terms of convergence speed?
- The PS aggregation method is widely adoptedCho et al. (2023); Chen et al. (2024a;b), but can it really guarantee convergence for the global model? What is the difference between SP and PS aggregation methods in terms of convergence speed?

- For more general aggregation algorithms, under what conditions can they guarantee the convergence of the global model?

Addressing these questions is of both theoretical and practical significance. From a theoretical standpoint, a unified convergence analysis helps us understand the fundamental principles that govern the success or failure of various LoRA aggregation strategies. It also provides a rigorous framework for comparing different methods on an equal footing. From a practical perspective, identifying the conditions that guarantee convergence can guide the design of more effective aggregation algorithms, enabling faster training and better performance in real-world FL scenarios.

**Contribution** Our research is primarily motivated by addressing the above questions, and on this basis, we have proposed some more general conclusions. We summarize our contributions as follows:

- We formally define the Aggregation-Broadcast Operator. Under mild assumptions, we establish both weak and strong convergence conditions. We prove that when weak convergence conditions are satisfied, the Aggregated Broadcasting Operator (ABO) ensures convergence of local models in the LoRA subspace at a rate of $O(1/\sqrt{T})$. When strong convergence conditions are met, it guarantees convergence of global models in the same subspace at the same rate.
- Especially, we prove that the SP Aggregation-Broadcast Operator satisfies the weak convergence condition but cannot achieve the optimal convergence rate due to broadcast errors, whereas the PS operator satisfies the strong convergence condition and achieves both global convergence and the optimal convergence rate.
- We perform comprehensive empirical studies to validate our theoretical findings. In particular, we investigate the effects of LoRA rank and the number of local training epochs on the convergence behavior of PS and SP aggregation methods, demonstrating strong consistency with our analytical predictions.

## 2 RELATED WORK

**Federated Learning** McMahan et al. introduced FedAvg in McMahan et al. (2017) as a decentralized and privacy-aware model training approach. Since then, numerous works have addressed the challenges of non-IID data Zhao et al. (2018), communication efficiency Konečný et al. (2016), and personalization Smith et al. (2017). Several algorithms have been proposed to improve optimization in heterogeneous settings, including FedProx Li et al. (2020a), SCAFFOLD Karimireddy et al. (2020), and MOON Li et al. (2021). Recent efforts also explore fairness Li et al. (2019) and adaptive aggregation Wang et al. (2020) to balance performance across clients.

**LoRA** Low-Rank Adaptation (LoRA) has emerged as an efficient Parameter-Efficient-Fine-Tuning (PEFT) method for Large Language Models (LLMs)Hu et al. (2022b). Early work, such as Universal Language Model Fine-tuning (ULMFiT), also explored efficient adaptation methods to reduce overhead Howard & Ruder (2018). Based on this, QA-LoRA Xu et al. (2024) introduces quantization-aware adaptation by integrating LoRA with low-bit quantization to further reduce memory usage. Similarly, QLoRA Dettmers et al. (2024) extends this idea by using 4-bit quantized LoRA adapters, achieving competitive performance with significantly lower memory footprint. In vanilla LoRA, the model rank requires manual configuration. To solve this issue, AdaLoRA Zhang et al. (2023) and AutoLoRAZhang et al. (2024c) automate the rank selection process, allowing the model to adaptively allocate parameters where most needed. In theoretical analysis, Sadhika et al.Malladi et al. (2023) investigate LoRA fine-tuning through the lens of kernel theory and show that, in the lazy training regime, its behavior closely mirrors that of full fine-tuning. Moreover, Zhu et al. (2024) shows that tuning LoRA matrix B is more impactful than tuning LoRA matrix A. Zeng et al.Zeng & Lee (2024) provide a theoretical analysis of LoRA's expressive power for both Fully Connected Neural Networks (FNNs) and Transformer Networks (TNs).

**LoRA-based Federated Learning** Based on the above advancements, numerous LoRA-based methods have been proposed for FL Wu et al. (2024); Cho et al. (2024); Yi et al. (2023); Bai et al.

(2024); Chen et al. (2024a); Zhang et al. (2024b); Guo et al. (2025). These approaches leverage low-rank LoRA adapters in place of full-rank local models to significantly reduce communication and computational overhead, while maintaining strong adaptability in heterogeneous environments.

# 3 AGGREGATION-BROADCAST OPERATOR

**Notation**   Let $W_i^{(t)}$, $B_i^{(t)}$, and $A_i^{(t)}$ denote the local model parameters and the LoRA adapter matrices for client $i$ at step $t$ (where $1 \leq i \leq m, 1 \leq t \leq T$). Correspondingly, let $W^{(t)}$, $B^{(t)}$, and $A^{(t)}$ represent the global model parameters at step $t$. The initial model $W_0$ refers to the pretrained global model weight. We define the set of global synchronization steps $\mathcal{I}_E$ as:

$$\mathcal{I}_E = \{nE \mid n \in \mathbb{N}^+\}, \tag{5}$$

where $E$ denotes the communication interval. If $t + 1 \in \mathcal{I}_E$, then step $t + 1$ corresponds to a communication round. Let $\mathcal{L}_i(W_i^{(t)}; \xi_{i,t})$ be the local loss function for client $i$ at step $t$ with $\mathbb{E}\left[\mathcal{L}_i(W_i^{(t)}; \xi_{i,t})\right] = \mathcal{L}_i(W_i^{(t)})$, where $\xi_{i,t}$ is sampled from the $i$-th client's local data uniformly at random at the training step $t$. Define the global objective as the weighted sum of local losses:

$$\mathcal{L}(W^{(t)}; \xi) = \frac{1}{m} \sum_{i=1}^{m} \mathcal{L}_i(W^{(t)}; \xi_{i,t}). \tag{6}$$

To describe the aggregation and broadcast process more generally, we define the Aggregation-Broadcast Operator:

**Definition 1** (**Aggregation-Broadcast Operator, ABO**). *The Aggregation-Broadcast Operator of the LoRA matrices $B_i^{(t+1)}$ and $A_i^{(t+1)}$ is defined by:*

$$\mathcal{P}(A_{1 \leq j \leq m}^{(t+1)}, B_{1 \leq j \leq m}^{(t+1)}) := \mathcal{P}(A_1^{(t+1)}, \cdots, A_m^{(t+1)}, B_1^{(t+1)}, \cdots, B_m^{(t+1)})$$
$$\mathcal{Q}(A_{1 \leq j \leq m}^{(t+1)}, B_{1 \leq j \leq m}^{(t+1)}) := \mathcal{Q}(A_1^{(t+1)}, \cdots, A_m^{(t+1)}, B_1^{(t+1)}, \cdots, B_m^{(t+1)})$$

*If $t + 1 \in \mathcal{I}_E$, then a communication round is triggered, and each client enters the aggregation-broadcast phase. During this phase, the local LoRA matrices $B_i^{(t+1)}$ and $A_i^{(t+1)}$ is updated via Aggregate-Broadcast Operators $\mathcal{P}$ and $\mathcal{Q}$, such that:*

$$B_i^{(t+1)} \leftarrow \mathcal{P}(A_{1 \leq j \leq m}^{(t+1)}, B_{1 \leq j \leq m}^{(t+1)})$$
$$A_i^{(t+1)} \leftarrow \mathcal{Q}(A_{1 \leq j \leq m}^{(t+1)}, B_{1 \leq j \leq m}^{(t+1)})$$

*for $1 \leq i \leq m$.*

According to this definition, we establish both SP and PS-Type Aggregation-Broadcast here:

**SP-Type Aggregation-Broadcast.**   When FL adopts the SP aggregation, the server first aggregates the global model based on local LoRA adapters $A_i$ and $B_i$ by using $\Delta W = \frac{1}{m} \sum_{i=1}^{m} B_i A_i$. After aggregation, the server applies SVD decomposition of $\frac{1}{m} \sum_{i=1}^{m} B_i A_i$ to $\tilde{U} \Sigma \tilde{V}^\top$, and broadcasts the result to all clients. In this process, the updated local LoRA matrices after the aggregation-broadcast step can be expressed as:

$$B_i^{(t+1)} \leftarrow \mathcal{P}(A_{1 \leq j \leq m}^{(t+1)}, B_{1 \leq j \leq m}^{(t+1)}) = \tilde{U}[:, :r] \Sigma[: r, :r] \tag{7}$$
$$A_i^{(t+1)} \leftarrow \mathcal{Q}(A_{1 \leq j \leq m}^{(t+1)}, B_{1 \leq j \leq m}^{(t+1)}) = \tilde{V}^\top[: r, :] \tag{8}$$

**PS-Type Aggregation-Broadcast.**   The PS method separately averages $B_i^{(t+1)}$ and $A_i^{(t+1)}$, which means that the global model aggregates by using $\Delta W = (\frac{1}{m} \sum_{i=1}^{m} B_i)(\frac{1}{m} \sum_{i=1}^{m} A_i)$, then broad-

casting the mean of each:

$$B_i^{(t+1)} \leftarrow \mathcal{P}(A_{1 \le j \le m}^{(t+1)}, B_{1 \le j \le m}^{(t+1)}) = \frac{1}{m} \sum_{i=1}^{m} B_i^{(t+1)} \tag{9}$$

$$A_i^{(t+1)} \leftarrow \mathcal{Q}(A_{1 \le j \le m}^{(t+1)}, B_{1 \le j \le m}^{(t+1)}) = \frac{1}{m} \sum_{i=1}^{m} A_i^{(t+1)} \tag{10}$$

The algorithm of FL with both SP and PS-Type Aggregation-Broadcast can be seen in Appendix. A.2 . Furthermore, the one-step update for the local LoRA matrices at round $t + 1$ by such ABO $\mathcal{P}$ and $\mathcal{Q}$ can be described as follows:

$$\begin{pmatrix} B_i^{(t)} \\ A_i^{(t)} \end{pmatrix} \xrightarrow{\text{local update}} \begin{pmatrix} B_i^{(t+1)} = B_i^{(t)} - \eta \nabla_B \mathcal{L}_i(W_i^{(t)}; \xi_{i,t}) \\ A_i^{(t+1)} = A_i^{(t)} - \eta \nabla_A \mathcal{L}_i(W_i^{(t)}; \xi_{i,t}) \end{pmatrix} \xrightarrow{\text{if } t+1 \in \mathcal{I}_E} \begin{pmatrix} \mathcal{P}(A_{1 \le j \le m}^{(t+1)}, B_{1 \le j \le m}^{(t+1)}) \\ \mathcal{Q}(A_{1 \le j \le m}^{(t+1)}, B_{1 \le j \le m}^{(t+1)}) \end{pmatrix}.$$
$$\tag{11}$$

for $1 \le i \le m$.

## 4 ANALYSIS

In this section, we establish convergence conditions and corresponding theorems for arbitrary Aggregation-Broadcast Operators under certain assumptions. Our analysis is primarily inspired by the work in Zhou & Cong (2017); Guo et al. (2025); Li et al. (2020b). However, unlike the study in Guo et al. (2025), this paper analyzes arbitrary Aggregation-Broadcast Operators under milder assumptions.

### 4.1 ASSUMPTION

To facilitate the theoretical analysis of LoRA-based aggregation in FL, we begin by introducing several standard assumptions that are widely used in the FL literature. These assumptions ensure that the local objective functions and model updates behave in a stable and analyzable manner. In particular, we assume the smoothness of the loss functions, uniform boundedness of their gradients, and uniform boundedness of the LoRA matrices during the model update process, which have been widely adopted in many theoretical analyses of FL Li et al. (2020b); Cho et al. (2021); Yu et al. (2019); Guo et al. (2025).

**Assumption 1.** $\mathcal{L}_1, \mathcal{L}_2, \cdots, \mathcal{L}_m$ *are all L-smooth. For all V and W,*

$$\|\nabla \mathcal{L}(V) - \nabla \mathcal{L}(W)\|_F \le L \|V - W\|_F$$

*It is equivalent to*

$$\mathcal{L}_i(V) \le \mathcal{L}_i(W) + \langle V - W, \nabla \mathcal{L}_i(W) \rangle + \frac{L}{2} \|V - W\|_F^2$$

**Assumption 2.** *The expected squared norm of the stochastic gradient is uniformly bounded, i.e.,* $\mathbb{E}\left[ \|\nabla \mathcal{L}_i(W_i^{(t)}; \xi_{i,t})\|^2 \right] = \|\nabla \mathcal{L}_i(W_i^{(t)})\|^2 \le G^2$, *for all* $i = 1, 2, \cdots, m$ *and* $t = 0, \cdots, T - 1$. *Here T denotes the total number of training steps for each client.*

**Assumption 3.** *Let* $W_i^{(t)} = W_0 + B_i^{(t)} A_i^{(t)}$. *There exist constants* $C_B > 0$ *and* $C_A > 0$ *such that* $\|B_i^{(t)}\|_F \le C_B$, $\|A_i^{(t)}\|_F \le C_A$ *for all* $i = 1, 2, \cdots, m$ *and* $t = 0, 1, \cdots, T - 1$.

### 4.2 CONVERGENCE RESULTS

We now discuss the convergence result for general Aggregate-Broadcast Operator(ABO). It's worth noting that: (1) we use the same learning rate $\eta$ during the whole training process and (2) all devices are active. Under these assumptions, we give the following convergence condition and convergence theorem respectively for arbitrary Aggregate-Broadcast Operator(ABO).

### 4.2.1 WEAK CONVERGENCE CONDITION

In what follows, we begin by introducing a convergence condition under which the local model is guaranteed to converge. We refer the condition as the **weak convergence condition**, which serves as the foundation for establishing convergence guarantee.

**Definition 2** (**Weak Convergence Condition**). *The Aggregation-Broadcast Operators(ABO) of LoRA matrices $B_i^{(t+1)}$ and $A_i^{(t+1)}$ are said to satisfy the Weak Convergence Condition if there exists a constant $R > 0$ such that:*

$$\mathbb{E}\left[\frac{1}{m}\sum_{i=1}^{m}\|\mathcal{P}(A_{1\leq j\leq m}^{(t+1)}, B_{1\leq j\leq m}^{(t+1)})\mathcal{Q}(A_{1\leq j\leq m}^{(t+1)}, B_{1\leq j\leq m}^{(t+1)}) - B_i^{(t+1)}A_i^{(t+1)}\|_F^2\right] \leq R^2\eta^2 \quad (12)$$

*for $1 \leq t \leq T$, where $\eta$ is learning the rate.*

Moreover, we show a sufficient condition of this definition as follows:

**Theorem 1** (**Sufficient Condition 1**). *The Aggregation-Broadcast Operators(ABO) of LoRA matrices $B_i^{(t+1)}$ and $A_i^{(t+1)}$ satisfy the Weak Convergence Condition if there exists a constant $R > 0$ such that:*

$$\mathbb{E}\left[\|\mathcal{P}(A_{1\leq j\leq m}^{(t+1)}, B_{1\leq j\leq m}^{(t+1)})\mathcal{Q}(A_{1\leq j\leq m}^{(t+1)}, B_{1\leq j\leq m}^{(t+1)}) - B_i^{(t+1)}A_i^{(t+1)}\|_F^2\right] \leq R^2\eta^2 \quad (13)$$

*for $1 \leq i \leq m$, $1 \leq t \leq T$, where $\eta$ is the learning rate.*

The details of this sufficient condition can be seen in Appendix. A.3.1. Based on the Weak Convergence Condition introduced above, we are now in a position to analyze the convergence behavior of the model under this setting. This is a general convergence theorem for arbitrary ABO that satisfy the Weak Convergence Condition.

**Theorem 2** (**Weak Convergence Theorem**). *Let Assumption 1, 2 and 3 hold. If the ABO satisfies the Weak Convergence Condition in Definition 2, the update for the local LoRA matrices follows Eq. (11). Then, for a learning rate $\eta > \xi > 0$ for some $\xi > 0$, the gradient of the local loss in expectation satisfies:*

$$\frac{1}{mT}\sum_{i=1}^{m}\sum_{t=1}^{T}(\mathbb{E}\left[\|\nabla_B L_i(W_i^{(t)})\|_F^2\right] + \mathbb{E}\left[\|\nabla_A L_i(W_i^{(t)})\|_F^2\right]) \leq \frac{D}{\eta T} + M\eta \quad (14)$$

*where $T$ is the total number of training steps for each client, $\mathcal{L}_i(W_i^0) - \mathcal{L}_i(W_i^*) \leq D$ for $\forall i$, $\frac{3}{2}L(\eta^2 C_A^2 C_B^2 G^4 + C_A^4 G^2 + C_B^2 G^2)\eta^2 + C_A C_B G^3 \eta^2 + \frac{L}{2}R^2\eta^2 + \frac{1}{2}(R^2 + G^2)\eta \leq M\eta^2$. Specifically, by choosing $\eta = \sqrt{\frac{D}{MT}}$, we obtain a convergence rate of $2\sqrt{\frac{DM}{T}}$.*

The proof of Theorem 2 is provided in the Appendix. A.5. Theorem 2 establishes that if the Aggregation-Broadcast Operator (ABO) satisfies the Weak Convergence Condition, the local model converges to a stationary point within the subspace spanned by $B$ and $A$, achieving a rate of $\mathcal{O}(1/\sqrt{T})$. Moreover, the convergence can be accelerated by reducing the values of $M$, which increase as $R$ increases. As a result, the upper bound $R$ in Definition 2, Eq. (12) directly affects the convergence speed: a larger $R$ leads to slower convergence, whereas a smaller $R$ results in faster convergence.

To further simplify the analysis, we ignore the dependency on the round $t$. It is known from the Weak Convergence Condition in Definition 2 that minimizing $R$ is equivalent to solving the following optimization problem:

$$\min_{\mathcal{P},\mathcal{Q}}\mathbb{E}\left[\frac{1}{m}\sum_{i=1}^{m}\|\mathcal{P}(A_{1\leq j\leq m}^{(t+1)}, B_{1\leq j\leq m}^{(t+1)})\mathcal{Q}(A_{1\leq j\leq m}^{(t+1)}, B_{1\leq j\leq m}^{(t+1)}) - B_i^{(t+1)}A_i^{(t+1)}\|_F^2\right] \quad (15)$$

By solving the optimal problem 15, we obtain:

**Corollary 1.** *The Aggregation-Broadcast Operators(ABO) $\mathcal{P}$ and $\mathcal{Q}$, can satisfy the Weak Convergence Condition and achieve the optimal convergence rate of the local model shown in Theorem 2 if the following equation holds:*

$$\mathcal{P}(A_{1\leq j\leq m}^{(t+1)}, B_{1\leq j\leq m}^{(t+1)})\mathcal{Q}(A_{1\leq j\leq m}^{(t+1)}, B_{1\leq j\leq m}^{(t+1)}) = \frac{1}{m}\sum_{i=1}^{m}B_i^{(t+1)}A_i^{(t+1)} \quad (16)$$

*for $1 \leq t \leq T$, moreover, we can get $R^2 = 8E^2G^2(C_A^4 + C_B^4)$.*

The proof of this corollary can be seen in Appendix. A.7. We refer to Eq. (16) as the **optimality condition** under the Weak Convergence Condition. It's obvious that the SP Aggregation Method (As shown in Eq. (3)) satisfies the Eq. (16) in Corollary 1 during the aggregation phase. However, issues arise during the broadcast phase. As shown in Eq. (7) and Eq. (8), which means that:

$$\mathcal{P}(A_{1\leq j\leq m}^{(t+1)}, B_{1\leq j\leq m}^{(t+1)})\mathcal{Q}(A_{1\leq j\leq m}^{(t+1)}, B_{1\leq j\leq m}^{(t+1)}) = \tilde{U}[:,:r]\Sigma[:r,:r]\tilde{V}^{\top}[:r,:] \neq \frac{1}{m}\sum_{i=1}^{m} B_i^{(t+1)} A_i^{(t+1)} \tag{17}$$

These results indicate that the SP aggregate-broadcast strategy cannot achieve the optimal convergence rate due to the broadcast error, also referred to as broadcast loss. Notably, this error becomes more significant as the LoRA rank $r$ decreases, thereby further slowing down the convergence of the global model. The convergence reaches its optimal rate only when the LoRA rank $r$ equals the rank of the optimal model. A more comprehensive analysis of this phenomenon will be presented in Section 5.2.

### 4.2.2 STRONG CONVERGENCE CONDITION

While the Weak Convergence Condition ensures the convergence of local models, it does not necessarily guarantee the convergence of the global model. In practice, the ultimate objective of FL is to achieve stable and efficient convergence of the global model. Therefore, we introduce a stronger convergence condition together with its corresponding theorem. This strong convergence condition imposes stricter requirements on the Aggregation-Broadcast Operator but provides the stronger guarantee that the global model will converge in the subspace spanned by the LoRA matrices $B$ and $A$.

**Definition 3** (**Strong Convergence Condition**). *The Aggregation-Broadcast Operators(ABO) of the LoRA matrices $B_i^{(t+1)}$ and $A_i^{(t+1)}$ are said to satisfy the Strong Convergence Condition if there exist constants $P > 0$ and $Q > 0$ such that:*

$$\mathbb{E}\left[\frac{1}{m}\sum_{i=1}^{m}\|\mathcal{P}(A_{1\leq j\leq m}^{(t+1)}, B_{1\leq j\leq m}^{(t+1)}) - B_i^{(t+1)}\|_F^2\right] \leq P^2\eta^2 \tag{18}$$

$$\mathbb{E}\left[\frac{1}{m}\sum_{i=1}^{m}\|\mathcal{Q}(A_{1\leq j\leq m}^{(t+1)}, B_{1\leq j\leq m}^{(t+1)}) - A_i^{(t+1)}\|_F^2\right] \leq Q^2\eta^2 \tag{19}$$

*for $1 \leq i \leq m$, $1 \leq t \leq T$.*

Similarly, we establish a sufficient condition for the Strong Convergence Condition:

**Theorem 3** (**Sufficient Condition 2**). *The Aggregation-Broadcast Operators(ABO) of the LoRA matrices $B_i^{(t+1)}$ and $A_i^{(t+1)}$ satisfy the Strong Convergence Condition if there exist some constant $P > 0$ and $Q > 0$ such that:*

$$\mathbb{E}\left[\|\mathcal{P}(A_{1\leq j\leq m}^{(t+1)}, B_{1\leq j\leq m}^{(t+1)}) - B_i^{(t+1)}\|_F^2\right] \leq P^2\eta^2 \tag{20}$$

$$\mathbb{E}\left[\|\mathcal{Q}(A_{1\leq j\leq m}^{(t+1)}, B_{1\leq j\leq m}^{(t+1)}) - A_i^{(t+1)}\|_F^2\right] \leq Q^2\eta^2 \tag{21}$$

*for $1 \leq i \leq m$, $1 \leq t \leq T$. Where $\eta$ is the learning rate.*

The details of this sufficient condition can be seen in Appendix. A.3.2. We define the global weight in step $t$ as $W^{(t)} = W_0 + \mathcal{P}(A_{1\leq j\leq m}^{(t)}, B_{1\leq j\leq m}^{(t)})\mathcal{Q}(A_{1\leq j\leq m}^{(t)}, B_{1\leq j\leq m}^{(t)})$. Then we obtain the following Strong Convergence Theorem under the Strong Convergence Condition.

**Theorem 4** (**Strong Convergence Theorem**). *Let Assumption 1, 2 and 3 hold. If the ABO satisfies the Strong Convergence Condition in Definition 3, the update for the local LoRA matrices is followed by Eq. (11). Then, for a learning rate $\eta > \xi > 0$ for some $\xi > 0$, the gradient of the global loss in expectation satisfy:*

$$\frac{1}{T}\sum_{t=1}^{T}(\mathbb{E}\left[\|\nabla_B\mathcal{L}(W^{(t)})\|_F^2\right] + \mathbb{E}\left[\|\nabla_A\mathcal{L}(W^{(t)})\|_F^2\right]) \leq 2(\frac{D}{\eta T} + (M+N)\eta) \tag{22}$$

*for $1 \leq i \leq m$, $1 \leq t \leq T$, where $\mathcal{L}_i(W_0) - \mathcal{L}_i(W_i^*) \leq D$, $R^2 = 4P^2Q^2\eta^2 + 3C_B^2Q^2 + 3C_A^2P^2$, $4G^2(Q^2 + P^2)\eta^2 + 4(C_A^2 + C_B^2)L^2R^2\eta^2 \leq 2N\eta$ and $\frac{3}{2}L(\eta^2C_A^2C_B^2G^4 + C_A^4G^2 + C_B^2G^2)\eta^2 + C_AC_BG^3\eta^2 + \frac{L}{2}R^2\eta^2 + \frac{1}{2}(R^2 + G^2)\eta \leq M\eta^2$. Specifically, by choosing $\eta = \sqrt{\frac{D}{(M+N)T}}$, we obtain a convergence rate of $4\sqrt{\frac{D(M+N)}{T}}$.*

The proof of this theorem is provided in Appendix. A.8. Similarly, if we ignore the dependency on $t$, minimize $R$ under Strong Convergence Condition in Definition 3 is equivalent to solve the following convex-optimization problemBoyd & Vandenberghe (2004):

$$\min_{\mathcal{P}} \mathbb{E}\left[\frac{1}{m}\sum_{i=1}^m \|\mathcal{P}(A_{1\leq j\leq m}^{(t+1)}, B_{1\leq j\leq m}^{(t+1)}) - B_i^{(t+1)}\|_F^2\right] \tag{23}$$

$$\min_{\mathcal{Q}} \mathbb{E}\left[\frac{1}{m}\sum_{i=1}^m \|\mathcal{Q}(A_{1\leq j\leq m}^{(t+1)}, B_{1\leq j\leq m}^{(t+1)}) - A_i^{(t+1)}\|_F^2\right] \tag{24}$$

By solving the optimization problem, we can get the corollary:

**Corollary 2.** *The Aggregation-Broadcast Operators(ABO) $\mathcal{P}$ and $\mathcal{Q}$ can satisfy the Strong Convergence Condition in Definition 3 and achieve the optimal convergence rate of the global model shown in Theorem 4 if the following equation holds:*

$$\mathcal{P}(A_{1\leq j\leq m}^{(t+1)}, B_{1\leq j\leq m}^{(t+1)}) = \frac{1}{m}\sum_{i=1}^m B_i^{(t+1)} \tag{25}$$

$$\mathcal{Q}(A_{1\leq j\leq m}^{(t+1)}, B_{1\leq j\leq m}^{(t+1)}) = \frac{1}{m}\sum_{i=1}^m A_i^{(t+1)} \tag{26}$$

*for $1 \leq t \leq T$, where $P^2 = 4E^2G^2C_A^4$, $Q^2 = 4E^2G^2C_B^2$, which lead to $R^2 = 64E^4G^4C_A^2C_B^2\eta^2 + 12E^2G^2(Q^2C_B^4 + P^2C_A^4)$.*

The proof of this corollary can be seen in Appendix. A.9. We refer to Eq. (25) and (26) as the **optimality condition** under the Strong Convergence Condition. It is worth noting that the PS Aggregation Method satisfies this optimality condition. As a result, the PS Aggregation Method is capable of achieving the optimal convergence rate of the global model if all clients share the same LoRA rank, which suggests that the PS aggregation method is relatively robust to the choice of LoRA rank. We will provide a more detailed analysis of this issue in Section 5.2.

## 5 EXPERIMENT AND EVALUATION

In this section, we introduce the experiments we conduct to verify our proposed theory.

### 5.1 EXPERIMENT SETUP

We verify the proposed convergence theory using a Multi-Layer Perceptron (MLP) on the MNIST, FMNIST, QMNIST and KMNIST datasets with 10 clients under a highly non-IID distribution, where each client holds all samples from only one class. We pick representative method FlexLoRA Bai et al. (2024) and RBLA Chen et al. (2024a) to represent SP and PS, respectively. All experiments are conducted with a fixed random seed 42.

- **Rank scale ratio** $\delta$**:** The LoRA rank of each layer scaled by $\delta$ as $\max([c \cdot \delta], 1)$ Chen et al. (2024a), where $c$ is a layer-specific constant. In our model, the based rank $c$ for each layer is set to 160, 160, and 100, respectively.
- **Total number steps** $T$**:** Throughout the training process, we specify the total number of steps trained by each client as $T$, where $T = \text{epoch} \cdot \text{rounds}$.

The following experiments are mainly designed to examine the sensitivity of PS-ABO and SP-ABO to different rank ratios and epochs. Based on our theoretical analysis, we can make the following observations:

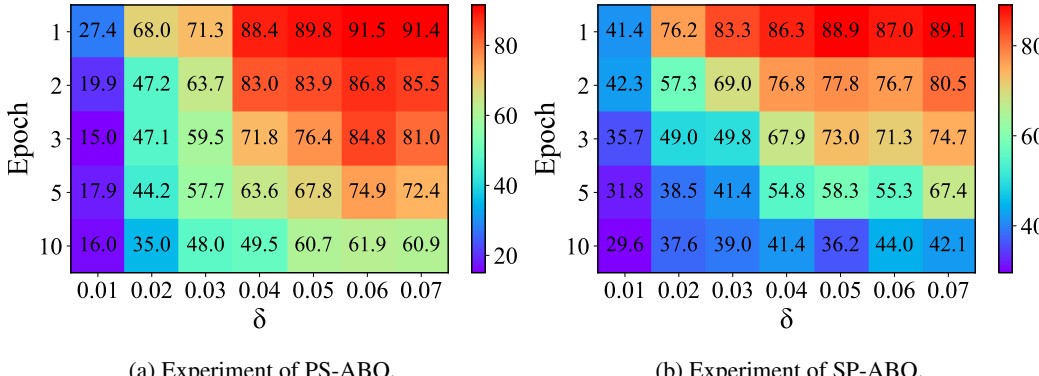

(a) Experiment of PS-ABO.                    (b) Experiment of SP-ABO.

Figure 1: The highest accuracies (%) of PS-ABO and SP-ABO at different rank ratios and epochs when the total number of steps $T = 300$ (Epoch * Round) on the MNIST dataset. Each cell represents the highest test accuracy the global model can reach in experiments with different $\delta$ and epoch.

**Different Rank.**   We know that the smaller the rank of a model, the weaker its expressive power tends to be. Consequently, as the rank decreases, the model's accuracy also drops. However, by Eq. 17, we know that due to the presence of broadcasting errors (caused by SVD), SP-ABO is more sensitive to the rank.

**Different Epoch.**   From Corollary 1 and Corollary 2 we know that $R_{sp}^2 = 8E^2G^2(C_A^4 + C_B^4)$, $R_{ps}^2 = 64E^4G^4C_A^2C_B^2\eta^2 + 12E^2G^2(Q^2C_B^4 + P^2C_A^4)$. These expressions indicate that, under a fixed total number of total steps $T$ (i.e. total number of epochs) and fixed random seed (which means the fixed $G$, $C_A$ and $C_B$), increasing the number of local epochs $E$ per communication round leads to a larger value of $R$, and thus slower convergence for both SP-ABO and PS-ABO. In other words, when $T$ is fixed, performing more local updates between communication rounds degrades overall convergence performance. Moreover, we observe that $R_{sp}^2 = \mathcal{O}(E^2)$ and $R_{ps}^2 = \mathcal{O}(E^4)$, which clearly shows that PS-ABO is more sensitive to the choice of epoch number.

## 5.2 EXPERIMENT RESULT

As shown in Fig. 1, for example, at epoch 1, as the rank decreases from 0.07 to 0.01, SP-ABO drops sharply from $91.4\%$ to $27.4\%$, whereas PS-ABO decreases more moderately from $89.1\%$ to $41.4\%$. A similar phenomenon can be observed across different epochs. This contrast demonstrates that SP-ABO is considerably more sensitive to rank reduction than PS-ABO. Moreover, when $\delta = 0.01$, as the epoch increases from 1 to 10, the accuracy of SP-ABO decreases by approximately 30 percentage points (from $91.4\%$ to $60.9\%$), whereas that of PS-ABO decreases by up to 47 percentage points (from $89.1\%$ to $42.1\%$). A similar trend is observed across different rank ratios.

Overall, these results illustrate that PS-ABO is more sensitive to the number of local epochs, while SP-ABO is more sensitive to the LoRA rank. These results are consistent with our theoretical analysis. Detailed experiment setup and additional experiments can be seen in Appendix. A.10.

## 6 CONCLUSION

In this paper, we presented a unified theoretical framework for analyzing the convergence behavior of LoRA-based FL. By introducing the concept of Aggregation-Broadcast Operators, we established a general convergence condition along with several sufficient conditions. Our framework not only provides convergence guarantees for the widely used SP-ABO and PS-ABO, but also offers insights into designing new aggregation methods with provable performance. Extensive experiments on standard benchmarks corroborate our theoretical findings. The findings contribute to a clearer understanding of LoRA in federated scenarios and may assist in developing more efficient and reliable model adaptation strategies.

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
