# OpenReview forum: "On the Convergence of LoRA-Based Federated Learning: A Unified Analysis of Aggregation-Broadcast Operators"
_ICLR.cc/2026/Conference — ICLR 2026 Conference Withdrawn Submission_

### Official Review · Reviewer_bujH · 2025-10-27

**Soundness:** 3
**Presentation:** 3
**Contribution:** 2
**Rating:** 4
**Confidence:** 5

**Summary:**

This paper provides the first unified convergence analysis of LoRA-based Federated Learning (FL) systems.
While LoRA significantly reduces communication costs by transmitting low-rank adapters instead of full model updates, how these low-rank matrices should be aggregated and broadcasted across clients remains underexplored.
The authors introduce a general theoretical framework based on the Aggregation–Broadcast Operator (ABO), which captures and unifies various existing LoRA aggregation schemes. Two categories are defined:
(1) SP-type (Sum–Product) aggregation, such as FedAvg and FlexLoRA; (2) PS-type (Product–Sum) aggregation, such as RBLA and FFA-LoRA. The paper provides convergence guarantees for both types: SP operators satisfy weak convergence (local convergence);
PS operators satisfy strong convergence (global convergence) with an optimal rate of O(1/T).
Extensive experiments on multiple benchmarks (MNIST, FMNIST, KMNIST, QMNIST) confirm the theoretical findings, showing the distinct trade-offs between rank reduction and local training epochs.

**Strengths:**

* The presentation is clear, with clean notation and solid mathematical exposition.
* The experiments align well with theoretical predictions, reinforcing the soundness of the analysis.

**Weaknesses:**

* The experiments are only on small vision datasets (MNIST variants); results on larger-scale or real-world FL settings (e.g., language models or CIFAR) would better demonstrate generality.
* Although convergence conditions are clear theoretically, the paper provides limited guidance on how to choose between SP and PS schemes in practice.
* Similar low-rank decomposition or subspace parameterization techniques (though not explicitly named “LoRA”) have already been explored in federated settings for model compression and aggregation, for example [1][2]. The paper does not sufficiently discuss or compare with these existing studies, which weakens its claim of novelty.

[1] FedLMT: Tackling System Heterogeneity of Federated Learning via Low-Rank Model Training with Theoretical Guarantees. ICML 2024.
[2] The Panaceas for Improving Low-Rank Decomposition in Communication-Efficient Federated Learning. ICML 2025

**Questions:**

How does the proposed analysis differ from previous low-rank decomposition approaches in federated learning, such as [1][2] work on communication-efficient low-rank FL?

[1] FedLMT: Tackling System Heterogeneity of Federated Learning via Low-Rank Model Training with Theoretical Guarantees. ICML 2024.
[2] The Panaceas for Improving Low-Rank Decomposition in Communication-Efficient Federated Learning. ICML 2025

---

### Official Review · Reviewer_BnTq · 2025-10-31

**Soundness:** 2
**Presentation:** 2
**Contribution:** 2
**Rating:** 2
**Confidence:** 4

**Summary:**

The paper studies FL when clients fine‑tune only LoRA adapters. It formalizes a general Aggregation‑Broadcast Operator (ABO) that maps client LoRA factors $(B_i^{(t+1)}, A_i^{(t+1)})$ to broadcasted factors $(P(\cdot), Q(\cdot))$. Within this framework, the authors define *Weak* and *Strong* Convergence Conditions**, prove $O(1/\sqrt{T})$ convergence rates for local (weak) and global (strong) convergence.

Additionally, this paper positions “Sum‑Product” (SP, averaging full low‑rank updates then SVD‑factoring) as satisfying only the weak condition due to SVD truncation error, while “Product‑Sum” (PS, averaging $B$ and $A$ separately) satisfies the strong condition, and it empirically compares these two on MNIST‑family datasets under extreme non‑IID (1 class/client) with grid sweeps over LoRA rank and local epochs.
This work demonstrates that SP is more sensitive to small ranks, while PS is more sensitive to numerous local epochs, aligning with the theory.



---
### LLM usage disclosure (reviewer)
I used GPT‑5 to help polish and organize this review; I take full responsibility for the content.

**Strengths:**

1. The ABO abstraction cleanly covers common LoRA‑FL aggregators (SP/FedAvg‑style on $\Delta W$, PS with mean of $A$ and $B$, and others by choice of $P,Q$). This helps compare methods on equal footing.

2. The analysis isolates an aggregation‑broadcast mismatch term $R$ (Def. 2) and shows how it controls rates $D/(\eta T)+M\eta$ (Theorem 2) and $D/(\eta T)+(M+N)\eta$ (Theorem 4). The narrative that SVD truncation in SP inflates $R$ is intuitive and practically applicable.

3. Corollaries suggest that if you can realize $PQ=\frac1m\sum_i B_iA_i$, you minimize $R$ under the weak condition; if you can realize $P=\overline{B}, Q=\overline{A}$, you minimize $R$ under the strong condition.

4. Heatmaps over rank and local epochs in Fig. 1 provide a compact, readable check of sensitivities predicted by the bounds.

**Weaknesses:**

1. **Assumptions are strong and under‑motivated.**

- **Bounded adapter norms** (Assumption 3: $|B_i^{(t)}|_F\le C_B), (|A_i^{(t)}|_F\le C_A)$ are nontrivial to guarantee for SGD without explicit projection/regularization. The paper relies on them but does not propose mechanisms ensuring they hold in practice.

- **All clients active; constant step‑size; single global $\eta$.** The sensitivity to partial participation and step‑size schedules is not addressed, which limits practical relevance.

2. **Sufficient conditions are almost tautological.** Theorems 1 & 3 restate the weak/strong conditions at the per‑client level, but they don’t provide constructive, verifiable criteria that can be used to check a new aggregator. The interesting part is the corollaries, but those essentially say “choose $P, Q$ to equal the population averages,” which is obvious when dimensions permit.

3. **PS “optimality” assumes equal ranks and compatible shapes.** Corollary 2 states PS attains the strong‑condition optimum “if all clients share the same LoRA rank.” Some of the LoRA-FL literature features heterogeneous rank. This analysis does not formally cover these practical cases, and the experiments do not evaluate them.

4. **Empirical evaluation is too limited**

- Only small‑scale **MLP on MNIST/Fashion/Q/KMNIST**, 10 clients, class‑partitioned non‑IID. No CIFAR/Imagenet, no NLP tasks, no larger models, no heterogeneous ranks, no partial participation.

- Also, reporting highest test accuracy achieved per setting is optimistic, final accuracy or area‑under‑curve is more informative. There are no variances or CIs.

5. **Clarity issues and internal inconsistencies.**

- In Fig. 1, subfigure labels and the text appear swapped.

- In Sec. 5.2, the statement "when $\delta=0.01$..."  must be $\delta=0.07$ to match the heatmap.

- Notation drifts between Aggregated Broadcasting Operator and Aggregation‑Broadcast Operator.

6. **Limited novelty in rate statements.**
The $O(1/\sqrt{T})$ type results for nonconvex FL with local steps under smoothness/bounded variance assumptions are well‑known. The novelty is the way the ABO mismatch enters the constants. That’s interesting, but incremental without broader empirical validation.

**Questions:**

Please see the weakness section.

---

### Official Review · Reviewer_5AtY · 2025-11-01

**Soundness:** 2
**Presentation:** 2
**Contribution:** 2
**Rating:** 2
**Confidence:** 3

**Summary:**

The paper introduces a unified convergence theory for LoRA-based federated learning through the Aggregation–Broadcast Operator (ABO) framework, which generalizes how clients’ LoRA updates are aggregated and redistributed. It analyzes two main types: Sum–Product (SP) which average the products $B_i A_i$ and then factorize via SVD and Product–Sum (PS) which average the factors separately and multiply $\bar{B} \bar{A}$. The paper also introduces weak and strong Convergence Conditions.

**Strengths:**

The ABO formulation is simple and clarifying; it makes precise what “aggregate then factorize” (SP) and “factorize then aggregate” (PS) are doing and offers general conditions under which either will converge.

**Weaknesses:**

-	The experiments use small MLPs on MNIST family datasets with 10 clients, whereas the motivation centers on large models where LoRA matters most. There is no evaluation on CIFAR like vision tasks, transformers, or LLM fine-tuning settings used by state-of-the-art LoRA FL baselines (e.g., FLoRA, FlexLoRA).

-	Fig.1 appears to contradict the intended conclusions. For epoch = 1, the PS panel shows accuracy rising from 27.4% at δ=0.01 to 91.4% at δ=0.07 (range ≈ 64 points), while the SP panel rises from 41.4% to 89.1% (range ≈ 47.7). Thus, PS looks more sensitive to rank than SP, contrary to the text’s claim that “SP is more sensitive to the LoRA rank.” Section_5.2 also swaps SP and PS numbers (e.g., attributing 27.4% to SP when it belongs to PS). This weakens the empirical support for the theoretical sensitivity statements.

-	The paper reports the best accuracy achieved over training rather than convergence curves, gradient norms, or communication/compute metrics tied to the theory (e.g., how E impacts the constants R,P,Q)

-	Methods that specifically target heterogeneous ranks and aggregation noise FLoRA (stacking), RBLA (rank aware), FFA LoRA (freeze one factor), and RoLoRA (alternating factors) are not compared experimentally, though they are among the most relevant competitors and are explicitly discussed in the related work.

-	The ABO analysis does not explicitly map to stacking (FLoRA) or alternating (RoLoRA) schemes; it is unclear whether those operators satisfy the weak/strong conditions and with what constants.

-	The main theorems assume all clients are active every round and homogeneous ranks for the PS optimality statement. Modern FL frequently uses partial participation and rank heterogeneity; some cited methods are designed around that. The analysis does not cover these cases.

-	The “optimality” claims are about constants, not rates (both results are $O(1/\sqrt{T})$. However, the optimal point in Corollary 1 minimizes $∑_i∥Z−BiAi∥$ at $Z=1/m(∑BiAi)$ , without a rank constraint. In practical SP, the server must broadcast rank r factors; the best rank-r solution to that least-squares problem is the truncated SVD of the mean, which the paper labels as “broadcast error” and uses to argue sub-optimality. That comparison is unfair unless the same rank-r constraint is enforced in the definition of “optimal” under Weak Convergence Condition. Clarify the objective being optimized (unconstrained vs rank-constrained).

**Questions:**

1.	In section_5.2, the text attributes values that visually belong to the other panel (e.g., 27.4% appears in the PS panel, not SP). Which panel corresponds to which method, and do the sensitivity conclusions still hold after correcting these swaps?

2.	Corollary 2 mentions PS achieves the optimal rate when clients share the same rank. How does the analysis change for heterogeneous ranks (the more common case)? Can zero padding or RBLA-style rank-aware averaging be brought under your strong condition, and what constants result?
3.	Can FLoRA style stacking or RoLoRA’s alternating updates be expressed as ABOs that satisfy weak/strong conditions? If so, what are the bounds?
4.	SP requires SVD; PS requires broadcasting full factor averages. What is the server-side compute and communication per round for SP vs. PS in your implementation, and how does this scale with model size?
5.	Under the Weak convergence condition, if the server must broadcast rank r factors, is the true optimal P, Q (minimizing your bound) the best rank r approximation to $1/m∑_iB_iA_i$  (i.e., truncated SVD)? If so, how does your “broadcast loss” discussion change? Please formalize the rank constrained counterpart of Corollary 1.

6. PS “optimality.” Is “optimal” meant only within the ABO discrepancy metric (minimizing R,P, Q)? Can you reconcile cases where minimizing R may not correlate with downstream accuracy (e.g., PS’s product of means bias vs. task loss)?

---

### Official Review · Reviewer_eCQc · 2025-11-02

**Soundness:** 2
**Presentation:** 3
**Contribution:** 2
**Rating:** 4
**Confidence:** 3

**Summary:**

This paper proposes a unified theoretical framework for analyzing the convergence behavior of Federated LoRA for both two classic aggregation methods, i.e., sum product-aggregation broadcast operator (SP-ABO) and sum product (PS)-ABO. This paper reveals that SP cannot achieve the optimal convergence rate due to broadcast errors, whereas the PS operator satisfies the strong convergence condition and achieves both global convergence and the optimal convergence rate. Extensive experiments evaluate the theoretical findings. However, I have some concerns as follows: 1) The recent work claims that PS has the aggregation error, i.e., FLoRA: Federated Fine-Tuning Large Language Models with Heterogeneous Low-Rank Adaptations. 2) The scenario of heterogeneous ranks among clients should also be considered in the theoretical analysis, but it is missing in this paper; 3) The experimental results are not complete. Although this paper focuses on the convergence analysis of SP and PS methods, state-of-the-art aggregation methods are also required in the experiments.

**Strengths:**

This paper proposes a unified theoretical framework for analyzing the convergence behavior of Federated LoRA for both two classic aggregation methods, i.e., sum product-aggregation broadcast operator (SP-ABO) and sum product (PS)-ABO. This paper reveals that SP cannot achieve the optimal convergence rate due to broadcast errors, whereas the PS operator satisfies the strong convergence condition and achieves both global convergence and the optimal convergence rate. Extensive experiments evaluate the theoretical findings. Overall, this paper aims to address a significant issue and well-written.

**Weaknesses:**

I have some concerns as follows: 1) The recent work claims that PS has the aggregation error, i.e., FLoRA: Federated Fine-Tuning Large Language Models with Heterogeneous Low-Rank Adaptations. 2) The scenario of heterogeneous ranks among clients should also be considered in the theoretical analysis, but it is missing in this paper; 3) The experimental results are not complete. Although this paper focuses on the convergence analysis of SP and PS methods, state-of-the-art aggregation methods are also required in the experiments.

**Questions:**

My questions are shown as follows:
1) The recent work claims that PS has the aggregation error, i.e., FLoRA: Federated Fine-Tuning Large Language Models with Heterogeneous Low-Rank Adaptations;
2) The scenario of heterogeneous ranks among clients should also be considered in the theoretical analysis, but it is missing in this paper;
3) The experimental results are not complete. Although this paper focuses on the convergence analysis of SP and PS methods, state-of-the-art aggregation methods are also required in the experiments.

---

### Note · Authors · 2025-11-21

I have read and agree with the venue's withdrawal policy on behalf of myself and my co-authors.